# A *Listeria monocytogenes*-Based Vaccine Formulation Reduces Vertical Transmission and Leads to Enhanced Pup Survival in a Pregnant Neosporosis Mouse Model

**DOI:** 10.3390/vaccines9121400

**Published:** 2021-11-26

**Authors:** Dennis Imhof, William Robert Pownall, Camille Monney, Anna Oevermann, Andrew Hemphill

**Affiliations:** 1Institute of Parasitology, Vetsuisse Faculty, University of Bern, Länggassstrasse 122, 3012 Bern, Switzerland; dennis.imhof@vetsuisse.unibe.ch; 2Graduate School for Cellular and Biomedical Sciences, University of Bern, Mittelstrasse 43, 3012 Bern, Switzerland; william.pownall@vetsuisse.unibe.ch; 3Department of Surgery, Small Animal Clinic, Vetsuisse Faculty, University of Bern, Länggassstrasse 128, 3012 Bern, Switzerland; 4Division of Neurological Sciences, DCR-VPH, Vetsuisse Faculty, University of Bern, Bremgartenstrasse 109A, 3012 Bern, Switzerland; camille.monney@vetsuisse.unibe.ch (C.M.); anna.oevermann@vetsuisse.unibe.ch (A.O.)

**Keywords:** *Neospora caninum*, *Listeria monocytogenes*, vaccine, vertical transmission, abortion, neosporosis mouse model

## Abstract

The apicomplexan parasite *Neospora caninum* is the worldwide leading cause of abortion and stillbirth in cattle. An attenuated mutant *Listeria monocytogenes* strain (Lm3Dx) was engineered by deleting the virulence genes *actA*, *inlA*, and *inlB* in order to avoid systemic infection and to target the vector to antigen-presenting cells (APCs). Insertion of *sag1*, coding for the major surface protein NcSAG1 of *N. caninum*, yielded the vaccine strain Lm3Dx_NcSAG1. The efficacy of Lm3Dx_NcSAG1 was assessed by inoculating 1 × 10^5^, 1 × 10^6^, or 1 × 10^7^ CFU of Lm3Dx_NcSAG1 into female BALB/c mice by intramuscular injection three times at two-week intervals, and subsequent challenge with 1 × 10^5^
*N. caninum* tachyzoites of the highly virulent NcSpain-7 strain on day 7 of pregnancy. Dose-dependent protective effects were seen, with a postnatal offspring survival rate of 67% in the group treated with 1 × 10^7^ CFU of Lm3Dx_NcSAG1 compared to 5% survival in the non-vaccinated control group. At euthanasia (25 days post-partum), IgG antibody titers were significantly decreased in the groups receiving the two higher doses and cytokines recall responses in splenocyte culture supernatants (IFN-γ, IL-4, and IL-10) were increased in the vaccinated groups. Thus, Lm3Dx_NcSAG1 induces immune-protective effects associated with a balanced Th1/Th2 response in a pregnant neosporosis mouse model and should be further assessed in ruminant models.

## 1. Introduction

*Neospora caninum* is a cyst-forming, intracellular apicomplexan parasite closely related to *Toxoplasma gondii*, causing significant diseases in farm animals. *N. caninum* is an economically significant parasite, especially in cattle, due to its ability to persist within an infected animal, causing repeated abortions and stillbirth [1]. Additionally, infection of *N. caninum* in cattle frequently leads to the birth of persistently infected calves, which can transmit the parasite to the next generation. While no cases of human infection with *N. caninum* have been registered yet, neosporosis-induced abortion and diseases cause major economic losses in the cattle industry and are, thus, an important veterinary health problem [2,3]. *N. caninum* exhibits a life cycle with three major infectious stages: (I) sporozoites encapsulated in oocysts that are generated in their definitive host and shed into the environment via feces; (II) rapidly proliferating tachyzoites, that cause acute disease in the intermediate host and are vertically transmitted to fetuses, which may lead to abortion or disease; (III) slowly dividing bradyzoites, which form tissue cysts, that can persist for years in the intermediate host without inducing any clinical symptoms [1]. Intermediate hosts become infected by oral uptake of oocysts or by ingestion of tissue cysts. In an immunocompetent animal, immunity and possibly other physiological responses induce the conversion from tachyzoites to bradyzoites, and cysts are formed, which are mostly found in the brain without apparent clinical signs. Nevertheless, the inherent immunomodulation that takes place during pregnancy triggers recrudescence and reconversion to tachyzoites, and vertical transmission can lead to abortion and stillbirth [1].

An efficient vaccine against neosporosis should fulfil the following criteria: (I) prevention of tachyzoite proliferation in pregnant cattle and other ruminants to reduce vertical transmission to the fetus; (II) reduction or prevention of oocysts shedding of definitive hosts to minimize horizontal transmission; (III) prevention of tissue cysts formation in intermediate hosts that have been infected with oocysts or tissue cysts. Therefore, a vaccine that stimulates the cellular immune response as well as systemic antibody response should be aimed for [4,5,6,7]. To date, there is no commercial neosporosis vaccine on the market. Several vaccine types have been investigated in murine and bovine animal models [4]. These include live-vaccines composed of low-virulence or genetically modified *N. caninum* strains. Live vaccines showed very promising results regarding efficacy but there are also disadvantages based on safety issues and production costs, which has hindered their further development [8]. Therefore, alternative vaccine strategies include DNA or subunit vaccines composed of parasite antigens involved in host cell interaction as vaccine targets are under investigation [4,8]. Other vaccines employed recombinant antigen cocktails fused to OprI to induce a balanced Th1/Th2 immune response against *N. caninum* [9], or chimeric antigens such as recNcMIC3-1-R consisting of immunogenic epitopes of NcMIC3, NcMIC1 and NcROP2, which induced a Th2-type immune response in pregnant mice infected with *N. caninum* [10]. While most of these recombinant subunit vaccines have yielded promising results in non-pregnant infection models, only few have been able to maintain a protective immune response during pregnancy, showing that the pregnancy-associated immunomodulation represents a major obstacle.

Over the last years, *Listeria monocytogenes* (*Lm*) has attracted considerable attention as a vaccine vector due to its ability to generate strong innate and adaptive immune responses during infection, and thus it is considered as a potent vaccine vector for tumor immunotherapy and for the prevention of infectious diseases [11,12,13]. Several virulence factors are responsible for the regulation of host cell invasion and intercellular spread of the bacterium. The two bacterial surface proteins InlA and InlB are mediating the invasion of almost all cells by binding to cellular receptors of the host cells, which induce bacterial endocytosis [14,15]. After endocytosis, the actin-assembly inducing protein ActA of *Lm* is essential to hijack the cellular actin network that allows the bacterium to spread from cell to cell. Nevertheless, *Lm* does not rely on InlA and InlB to be phagocytosed by professional phagocytes including antigen-presenting cells (APCs). Following phagocytosis, *Lm* is either destroyed in the phagolysosome or escapes from phagosome into the cytosol [15]. Once in the cytosol, *Lm* secretes proteins that are degraded by proteasomes and peptides are loaded on major histocompatibility complex (MHC) class I for CD8^+^ T cell presentation [11,16]. Bacterial proteins that are degraded within the phagolysosome are delivered to MHC class II molecules for CD4^+^ T cell presentation. CD4^+^ T cells further differentiate into Th1 or Th2 helper cells [13].

In a previous study, we have shown that virulence of a newly developed *Lm* vaccine vector, Lm3Dx, was significantly attenuated by deleting the three important virulence genes *actA*, *inlA* and *inlB* to prevent systemic infection. However, Lm3Dx maintained immunogenicity as described previously [17]. Additionally, the fosfomycin resistance gene was deleted resulting in a fosfomycin sensitive vaccine vector. The gene coding for the immunodominant *N. caninum* surface antigen NcSAG1 was inserted into the attenuated mutant vaccine vector Lm3Dx, resulting in the Lm3Dx_NcSAG1 vaccine vector that elicited a Th1-biased immune response [17]. In this study, we report on a challenge experiment to demonstrate proof-of-principle for the efficacy of Lm3Dx_NcSAG1 in a standardized pregnant neosporosis mouse model. Lm3Dx_NcSAG1 was applied to mice three times at two-week intervals with the latest dose administered during pregnancy. These mice were then challenged with the highly virulent *N. caninum* NcSpain-7 strain to assess the efficacy of the mutant *Lm*-vaccine.

## 2. Materials and Methods

### 2.1. Host Cells, Parasite and Primers

If not mentioned otherwise, all cell culture media were purchased from Gibco-BRL (Zürich, Switzerland). Human foreskin fibroblasts (HFF; ATCC^®^ SCRC-1041^TM^) were maintained in Dulbecco’s modified Eagles’ medium (DMEM) supplemented with 10% FCS and 1% antibiotics/antimycotics at 37 °C, 5% CO_2_ in T25, T75 or T125 cell culture flasks (Sarstedt, Sevelen, Switzerland). BALB/c dermal fibroblasts (CELLNTEC Advanced Cell systems AG, Bern, Switzerland) were maintained under the same conditions as described for HFF. The highly virulent *N. caninum* NcSpain-7 isolate was maintained in HFF as described before and transferred to BALB/c dermal fibroblasts prior to infection [18]. Primers and Probes for real-time (RT) qPCR were purchased from Mycrosynth (Balgach, Switzerland).

### 2.2. Generation and Quality Control of the Mutant Vaccine Strain Lm3Dx_NcSAG1

The attenuated mutant vaccine vector Lm3Dx_NcSAG1 was engineered as previously described [17]. Fosfomycin minimal inhibitory concentration (MIC) and genetic stability of Lm3Dx_NcSAG1 were determined as described earlier [17].

### 2.3. Ethics Statement

All animal protocols were approved by the Animal Welfare Committee of the Canton of Berne (License BE 113/19). Six-week-old female and male BALB/c mice were purchased from a commercial breeder (Charles River, Sulzberg, Germany) and were maintained in a common room under controlled temperature and a 14 h light/10 h dark cycle according to guidelines of the Animal Welfare Legislation of the Swiss Veterinary Office. Mice were housed in the facility for two weeks prior the start of the challenge study for acclimatization to the environmental conditions.

### 2.4. Efficacy Evaluation of the Mutant Vaccine Strain Lm3Dx_NcSAG1 in Non-Pregnant and Pregnant Mice Infected with NcSpain-7 Tachyzoites

A total of 80 female and 40 male BALB/c mice, eight weeks of age, were used for this vaccination study. After the arrival of mice, females were randomly distributed into five experimental groups (16 mice/group; 2 females/cage): 1 × 10^7^ CFU Lm3Dx_NcSAG1 + infection; 1 × 10^6^ CFU Lm3Dx_NcSAG1 + infection; 1 × 10^5^ CFU Lm3Dx_NcSAG1 + infection; 1 × 10^6^ CFU Lm3Dx (empty vector) + infection (positive control, C+); PBS + BALB/c dermal fibroblasts (negative control, C−). Immunization schedule, mating, blood, and organ sample collection as well as euthanasia are shown in Figure 1.

Mice were immunized three times at two-week intervals, either with Lm3Dx_NcSAG1, Lm3Dx (C+) or PBS (C−), by intramuscular injection in the thigh. Eight days post-second immunization, mice were oestrus-synchronized for three days by the Whitten effect [19] and one male was housed together with two females for 72 h. The third vaccination was applied after males were separated from females, four days prior to infection. Three days before challenge, NcSpain-7 tachyzoites were transferred from HFF monolayer cultures to BALB/c dermal fibroblasts and were maintained at 37 °C and 5% CO_2_ [9]. On day seven post mating, NcSpain-7 tachyzoites were collected from cell culture flasks and counted as described previously [20], and 1 × 10^5^ tachyzoites/mouse were injected subcutaneously. Mice of the negative control (C−) received BALB/c dermal fibroblasts only. After infection, mice were observed daily and were weighted every third or fourth day to confirm pregnancy and/or detect potential abortions. At day 18 post-mating, pregnant mice were transferred into single cages to rear their offspring in calm, while non-pregnant mice were maintained in groups of 3–4 animals. Pregnant mice gave birth between day 20 and 22. Data on clinical signs, fertility, litter size, neonatal, and postnatal mortality were recorded. At 25 days post-partum (p.p.), all mice and pups were euthanized in a chamber by isoflurane and CO_2_. Blood and brain samples were collected and stored at −20 °C. Additionally, samples of brain were formalin-fixed, paraffin embedded, cut at 4 µm and stained with hematoxylin and eosin (HE). The spleen of every non-pregnant mouse and dam was removed aseptically for splenocyte re-stimulation assays.

### 2.5. Analysis of Cerebral Parasite Burden by Real-Time (RT) qPCR and Histology

The cerebral parasite burden was quantified in non-pregnant mice, dams and surviving pups by RT-qPCR designed specifically for *N. caninum* [21,22]. DNA purification was conducted with the NucleoSpin DNA RapidLyse Kit (Macherey-Nagel, Oesingen, Switzerland) according to the manufacturer’s instruction. Concentration of DNA was quantified by using the QuantiFluor double-stranded DNA system (Promega, Madison, WI, USA). TaqMan probe-based RT-qPCR was performed in a CFX96 qPCR instrument (Bio-Rad Laboratories AG, Cressier, Switzerland) to quantify *N. caninum*. CFX manager software version 1.6 was used for the analysis of the PCR results. RT-qPCR is targeted to the repetitive genomic sequence NC5 of *N. caninum* [21]. The reaction mixture (10 µL per reaction) contains 5 µL of 2 × Mastermix (SensiFAST^TM^ Probe NO-ROX Kit; Bioline Meridian Lifescience, Memphis, TN, USA), 500 nM forward primer Np21plus (5′-CCCAGTGCGTCCAATCCTGTAAC-3′) and reverse primer Np6plus (5′-CTCGCCAGTCAACCTACGTCTTCT-3′) [21], 100 nM of detection probe NC5-1 (5′-*FAM*-CACGTATCCCACCTCTCACCGCTACCA-*BHQ-1*-3′) [22] and 5 ng of sample DNA. Additionally, 300 nM dUTP (supplementary to dTTP included in the 2 × Mastermix) and 1 unit of heat-labile Uracil DNA Glycosylase (UDG) (both from Bioline Meridian Lifesciences) were included in the reaction mixture to remove eventual carry-over contaminations from previous reactions, as described previously [23]. For UDG-mediated decontaminations, the temperature profile included an initial 10 min incubation at 40 °C followed by a 5 min denaturation period at 95 °C. Subsequently, DNA amplification was achieved during 50 cycles of 10 s at 95 °C and 30 s at 60 °C. After each cycle, light emission by the fluorophore was measured at 60 °C. Brain samples from non-pregnant and pregnant mice were tested in duplicates and pup brains were measured in single values. As external quantification standards, samples containing DNA equivalents from approximately 1000, 100, 10 *N. caninum* tachyzoites were included. Additionally, the parasite burden and associated lesions were assessed by microscopical analysis of HE stained brain sections.

### 2.6. Analysis of Humoral Immune Responses by Enzyme-Linked Immunosorbent Assay (ELISA)

IgG antibody titers against NcSAG1 and *N. caninum* crude extract were assessed by ELISA as described earlier [24,25]. Briefly, 96-well plates were coated with 100 ng/well recNcSAG1 or 200 ng/well NcSpain-7 soluble tachyzoite protein extract diluted in coating buffer (50 mM sodium bicarbonate; 50 mM sodium carbonate dissolved in ultrapure water; pH 9.6) per well. After overnight incubation at 4 °C, plates were washed three times with washing buffer (0.05% PBS-Tween-20) before non-specific binding sites were blocked with blocking buffer (1% bovine serum albumin in 0.05% PBS-Tween-20) for 2 h at room temperature. Then, 4-fold serial dilution of each serum sample was performed in blocking solution including positive (vaccinated with Lm3Dx and infected with *N. caninum*) and negative (treated with PBS and inoculated with dermal fibroblasts only) controls, before samples were applied and incubated during 90 min at room temperature. After three washes, secondary antibodies (goat anti-mouse-IgG conjugated to alkaline phosphatase (SouthernBiotech, Birmingham, AL, USA) diluted 1:2000 in blocking solution) were added for 1 h incubation at room temperature. Alkaline phosphatase substrate was added to develop the enzymatic reaction and absorbance was measured as optical density (OD) at 405 nm in a microplate reader (EnSpire^TM^ 2300 Multilabel Reader, PerkinElmer, Schwerzenbach, Switzerland). The same positive and negative serum samples were used for each plate to compare OD values between samples measured in different plates. OD values were converted into a relative index per cent (RIPC) value using the following formula [RIPC = (OD_405nm_ sample − OD_405nm_ negative control)/(OD_405nm_ positive control − OD_405nm_ negative control) × 100] [18].

### 2.7. Determination of Cytokine Levels in Splenocyte Culture Supernatants by Multiplex Immunoassay 

Spleens were collected after euthanasia (25 days p.p.) and splenocytes were isolated as described [18]. Cultured splenocytes were re-stimulated with concanavalin A (5 µg/mL; Sigma, St. Louis, MO, USA), with soluble *N. caninum* protein extract (20 µg/mL) or remained unstimulated as a negative control. After 72 h of stimulation, supernatants were collected and stored at −80 °C. Luminex xMAP technology was used for cytokine measurement [26]. Multiplexing analysis was conducted using the Luminex™ 200 system (Luminex, Austin, TX, USA) by Eve Technologies (Calgary, AB, Canada). Cytokine levels were measured using Eve Technologies’ mouse high sensitivity T cell discovery array assay (MilliporeSigma, Burlington, MA, USA) according to manufacturer’s instructions. The multiplex array contains IFN-γ, IL-4 and IL-10. Individual analyte sensitivity values are available in the MilliporeSigma MILLIPLEX^®^ MAP protocol.

### 2.8. Statistical Analysis

Statistical analysis of cerebral parasite burdens, IgG antibody titers and encephalitic grades were compared between groups by the non-parametric Kruskal–Wallis test, including Dunn’s multiple comparison correction. If statistical differences were detected, a Mann–Whitney-*U* test was subsequently applied comparing only two groups with each other. Neonatal and postnatal mortality rates, as well as vertical transmission rates, were analyzed between the vaccinated groups and the positive control group by Chi-square (and Fisher’s exact) test. The pup mortality time was compared by plotting survival events at each time point in Kaplan–Meier graphs and survival curves were compared by the Log-rank (Mantel-Cox) test. Statistical analysis was performed using Graphpad Prism version 9.2.0 for MacOSX (GraphPad Software, La Jolla, CA, USA, www.graphpad.com accessed on 3 June 2020).

## 3. Results

### 3.1. Safety and Efficacy of the Mutant Listeria Strain Lm3Dx_NcSAG1 in the Pregnant Neosporosis Mouse Model

The results of this study are summarized in Table 1. Safety specifications of the attenuated mutant *Listeria* strain Lm3Dx_NcSAG1 were evaluated and described in detail previously [17]. Reproductive parameters, such as litter size and fertility rates, were not influenced by the vaccine vector at the two highest vaccination doses (1 × 10^7^ CFU and 1 × 10^6^ CFU). At the lowest dose (1 × 10^5^ CFU), only four out of 16 mice became pregnant. The reason for this low fertility rate is attributable to the continuous fighting of males and females during mating in this group, for which the reasons remain unknown.

None of the dams which were vaccinated three times with Lm3Dx_NcSAG1 showed clinical signs of neosporosis until the end of the study. In contrast, four out of ten pregnant mice that were inoculated with the empty vector Lm3Dx (C+), exhibited moderate neurological symptoms, ruffled coat and slower movements.

Vaccination with the highest inoculation dose (1 × 10^7^ CFU Lm3Dx_NcSAG1) resulted in 67% pup survival, while in the intermediate (1 × 10^6^ CFU Lm3Dx_NcSAG1) and lowest dose (1 × 10^5^ CFU Lm3Dx_NcSAG1) 46% and 31% of the pups survived, respectively (Figure 2). In the positive control group, where mice had been inoculated with the empty vector Lm3Dx only, 59 out of 62 pups died until the end of the experiment, resulting in 95% postnatal mortality, while all pups of the negative control group survived (Figure 2).

Seven out of eight brain samples from dams vaccinated with 1 × 10^7^ CFU, five out of eight brains from dams vaccinated with 1 × 10^6^ CFU and two out of four brains from dams vaccinated with 1 × 10^5^ CFU were tested PCR-positive for *N. caninum*, while all brain samples in the positive control group were *N. caninum* PCR-positive (Table 1). The significant reduction of cerebral parasite burden was achieved in none of the vaccinated dam groups compared to the positive control (Figure 3A).

In non-pregnant mice, *N. caninum* DNA was detected in five out of eight (1 × 10^7^ CFU), in one out of eight (1 × 10^6^ CFU) and in six out of 11 (1 × 10^5^ CFU) brain samples. Additionally, five out of six brains were *N. caninum* PCR-positive in the positive control group and all brain samples in the negative control group were *N. caninum* PCR-negative (Table 1). The cerebral parasite burden of the two higher vaccination doses was significantly reduced compared to the C+ group (* *p* < 0.0193, *** *p* < 0.0007), while no significant reduction was measured between the lowest vaccination dose and the positive control (Figure 3B).

Vertical transmission was strongly inhibited in dams vaccinated with 1 × 10^7^ CFU Lm3Dx_NcSAG1, resulting in 61% *N. caninum* PCR-negative pups. 23% of pups were tested negative from dams vaccinated with the intermediate dose of Lm3Dx_NcSAG1 (1 × 10^6^ CFU) and only 12% of pups from dams vaccinated with the lowest dose (1 × 10^5^ CFU). Vertical transmission was almost 100% in the positive control group and no *N. caninum* DNA was detected in the negative control group (Table 1). 

In conclusion, vaccination with 1 × 10^7^ CFU Lm3Dx_NcSAG1 resulted in a clear inhibition of vertical transmission leading to a strongly increased pup survival. Additionally, vaccination with Lm3Dx_NcSAG1 led to a significant reduction of the cerebral parasite burden in non-pregnant mice, but not in dams.

### 3.2. Humoral and Cellular Immune Responses

Sera were collected from all dams and non-pregnant mice at the endpoint (38–40 days post challenge), and IgG responses were assessed by ELISA using plates coated with *N. caninum* extract (Figure 4A) and recombinant NcSAG1 (Figure 4B). Seroconversion was achieved in all adult mice (Table 1). In non-pregnant mice, no differences in IgG levels against *N. caninum* extracts were detected between vaccinated mice and the positive control. Whereas in dams, IgG antibody titers were significantly decreased at the two higher vaccination doses compared to the C+ group (* *p* < 0.0434 for 1 × 10^7^ CFU; * *p* < 0.0343 for 1 × 10^6^ CFU), but not between the lowest vaccination dose (1 × 10^5^ CFU) and the positive control (Figure 4A). No statistically significant differences in the intensity of the IgG responses against the recombinant NcSAG1 were detected between vaccinated and non-vaccinated animals from all treatment groups (Figure 4B).

To assess splenocyte recall responses of cultured spleen cells obtained from the different experimental groups, spleen cells were collected at the endpoint and cultures were stimulated with either concanavalin A (ConA), soluble *N. caninum* extract or remained unstimulated (Figure 5). Measurements of cytokines in culture supernatants stimulated with a *N. caninum* extract demonstrated a strong interferon-gamma (IFN-γ) response in non-pregnant mice and dams which were immunized with either Lm3Dx_NcSAG1 or Lm3Dx (empty vector) but not with PBS (Figure 5A). Similar results were obtained for interleukin-4 (IL-4) with highest levels in non-pregnant mice and dams vaccinated with 1 × 10^7^ CFU Lm3Dx_NcSAG1 and slightly decreased in mice inoculated with 1 × 10^6^ CFU Lm3Dx_NcSAG1 and 1 × 10^6^ CFU Lm3Dx (Figure 5B). In addition, high levels of the anti-inflammatory cytokine IL-10 were measured in supernatants of splenocyte cultures stimulated with *N. caninum* extract in non-pregnant and pregnant animals. In non-pregnant mice, the peak IL-10 expression was observed at the highest inoculation dose, while in dams, the highest IL-10 expression was observed at 1 × 10^6^ CFU Lm3Dx_NcSAG1 (Figure 5C).

### 3.3. Histology

Histology revealed encephalitic lesions consistent with *N. caninum* infection in 14/16 mice vaccinated with the empty vector Lm3Dx and in 5/16, 4/15 (one brain could not be processed for histological analysis) and 8/16 mice vaccinated with 1 × 10^5^ CFU, 1 × 10^6^ CFU and 1 × 10^7^ CFU of Lm3Dx_NcSAG1, respectively. Encephalitic lesions consisted of lymphohistiocytic meningitis and encephalitis with gliosis and sometimes distinct and large granulomas. Representative histological images of the encephalitis grades 0–3 are shown in Figure 6. 

Not only were the numbers of mice with encephalitic lesions clearly lower in the groups vaccinated with the NcSAG1 carrying vector, the severity grade of encephalitis was also lower in these groups compared to the groups vaccinated with the empty vector. The difference was significant in all groups vaccinated with Lm3Dx_NcSAG1 compared to the positive control group that received Lm3Dx (Figure 7).

## 4. Discussion

Despite considerable efforts, there is currently no effective vaccine against neosporosis on the market. Experimental live vaccines comprised of attenuated *N. caninum* strains exhibited promising efficacy in pregnant mice and cattle [27,28,29,30,31], while classical subunit vaccines composed of recombinant antigens showed rather limited efficacy [2,32,33]. However, the pharmaceutical industry appears reluctant to invest in a product with a limited shelf life, high production costs, and uncertainty about development of virulence [34]. One of the major obstacles in the development of a subunit vaccine against neosporosis has been the immunomodulation that takes place during pregnancy. Thus, many antigens and vaccine formulations have shown promising effects in non-pregnant animals, but most have not been able to maintain protectivity during pregnancy, which is classically associated with a switch from a Th1 to a Th2 biased immunity to prevent detrimental effects and ensure fetal development [2]. Thus, there is a clear need for novel preventive measures to counteract the considerable economic impact of neosporosis.

We here describe the efficacy of a novel live-vaccine vector, which is based on the obligate intracellular bacterium *Lm*. *Lm* has been described as a promising vaccine vector for the prevention of infectious diseases, as well as for cancer treatment, due to its ability to induce strong innate and adaptive immune responses [16]. Additionally, *Lm* can be genetically manipulated in an efficient and simple manner. Several virulence genes, namely *inlA*, *inlB* and *actA*, were deleted to generate the attenuated mutant vector Lm3Dx to avoid systemic infection and intercellular spread. The major immunodominant surface antigen of *N. caninum,* NcSAG1, was inserted into the *actA* locus, resulting in the Lm3Dx_NcSAG1 vaccine strain. To generate a fosfomycin sensitive strain, the fosfomycin resistance gene *fosX* was deleted. Fosfomycin sensitivity is useful as a phenotypic marker to discriminate between Lm3Dx and the natural *Lm* wild type. Moreover, fosfomycin may be used as an alternative antibiotic for the elimination of potentially remaining bacteria after vaccination [17].

The safety aspects of this novel attenuated vaccine strain have been evaluated in a previous study showing that, compared to the virulent wildtype parental strain, the attenuated vector Lm3Dx does not spread to internal organs, does not persist in non-lactating mice and is not shed to relevant numbers into the environment [17].

Based on fulfilling the promising safety requirements upon vaccination in mice, Lm3Dx_NcSAG1 was evaluated in this study for its efficacy in reducing brain infection in non-pregnant mice and preventing vertical transmission and enhancing pup survival in pregnant BALB/c mice experimentally infected with tachyzoites of the highly virulent NcSpain-7 strain [20]. While a plethora of potential neosporosis vaccine candidates have been evaluated in recent years [35,36], NcSAG1 was chosen as target antigen since it represents a highly immunodominant, surface protein of *N. caninum* tachyzoites, which is implicated in the host cell invasion process [37,38]. NcSAG1 applied as DNA vaccine and as recombinant antigen [39,40,41,42] had been shown earlier to confer partial protection against cerebral infection in non-pregnant mice.

In this study, vaccination of mice with Lm3Dx_NcSAG1 resulted in highly promising results in both pregnant and non-pregnant mouse models of neosporosis. The vaccine did not induce any adverse effects in terms of fertility. A dose-dependent effect was observed, with the two highest doses of 1 × 10^7^ and 1 × 10^6^ CFU significantly increasing pup survival and reducing vertical transmission. In addition, dams vaccinated with 1 × 10^7^ and 1 × 10^6^ CFU of Lm3Dx_NcSAG1 exhibited diminished IgG titers compared to dams vaccinated with Lm3Dx alone, which probably reflects a lower disseminated parasite load, while this effect was not seen in non-pregnant animals. Nevertheless, application of Lm3Dx_NcSAG1 resulted in reduced cerebral infection in non-pregnant mice, but not in dams, which is most likely due to the immunomodulation that occurs in pregnant animals [2].

We have shown earlier through cytokine measurements after splenocyte re-stimulation with NcSAG1 or *N. caninum* crude extract that Lm3Dx_NcSAG1 induced a strong cellular immune response with high IFN-γ production and moderate IL-5 secretion, indicating that the attenuated vaccine vector induced a Th1-biased immune response [17]. In this experiment, cytokine recall responses of splenocytes were harvested at the end of the experiment and were stimulated with *N. caninum* antigens, which produced increased levels of the Th1 marker IFN-γ, but also the Th2 cytokine IL-4, and the regulatory cytokine IL-10. Thus, at the endpoint of this experiment the mice had elicited a balanced Th1/Th2 response, which has been shown earlier, although by applying bacterially expressed recombinant antigens, to be an important prerequisite for immune protection during pregnancy [9,18]. In one of these studies, NcPDI, NcROP2 and NcROP40, all of which are involved in the physical interaction between *N. caninum* tachyzoites and host cells, were expressed as OprI-fused recombinant antigens and were used for vaccination in the neosporosis mouse model. Vaccination with the OprI antigen cocktail induced a balanced Th1/Th2 immune response resulting in reduced vertical transmission, and postnatal mortality in pups was inhibited by 35%. In addition, a significant protection against cerebral infection was measured in adult mice [9]. In this study, employing only NcSAG1, the protective effects on pups were much higher. Thus, further studies will focus on the simultaneous expression of several antigens within the same Lm3Dx vaccine strain in order to investigate whether this will induce an even higher degree of protection. In addition, it will be important to investigate the mechanism that leads to enhanced protection following vaccination.

Based on the results of this vaccine study, it appears that live vaccines are still the most effective vaccine candidates. However, in contrast to cultivation of potentially attenuated live vaccines based on parasites, the attenuated bacterial mutant vaccine strain Lm3Dx_NcSAG1 is much easier to cultivate, clearly fulfils multiple safety requirements in vitro and in vivo and induces a Th1-biased immunity against the target antigen that protects against infection [17]. In addition, Lm3Dx_NcSAG1 can be easily tailored by insertion of other vaccine candidate antigens into the free loci of *actA*, *InlA* or *InlB*, or insertion of only the immunogenic epitopes of potential vaccine candidates into the vector. Besides, its usefulness to induce immune protection against neosporosis, Lm3Dx represents a potentially versatile tool to target other related apicomplexan parasites, such as *T. gondii*, *B. besnoiti*, *Sarcocystis* or *Eimeria.*

In conclusion, we have shown here that the attenuated mutant vector Lm3Dx_NcSAG1 exhibited promising efficacy against congenital and cerebral neosporosis in the mouse model. Further investigations are underway to improve its features and to investigate its potential use for the prevention of neosporosis in farm animals, and potentially other parasitic diseases.

## Figures and Tables

**Figure 1 vaccines-09-01400-f001:**
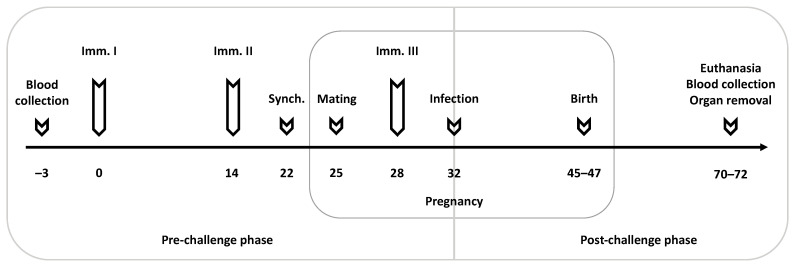
Timeline of the vaccination experiment. Mice were immunized three times at two-week intervals with the third immunization shortly after mating (Imm. 1–3). Eight days post-second immunization, mice were oestrus-synchronized (Synch.) before two females and one male were put together for mating. Seven days post mating, all mice, except of the negative control (C−), were challenged with a sublethal dose of 1 × 10^5^ NcSpain-7 tachyzoites. Mice were monitored daily for clinical signs, and neonatal and postnatal mortality rates were recorded. Non-pregnant mice, dams and pups were euthanized between day 70–72, corresponding to 25 days p.p.

**Figure 2 vaccines-09-01400-f002:**
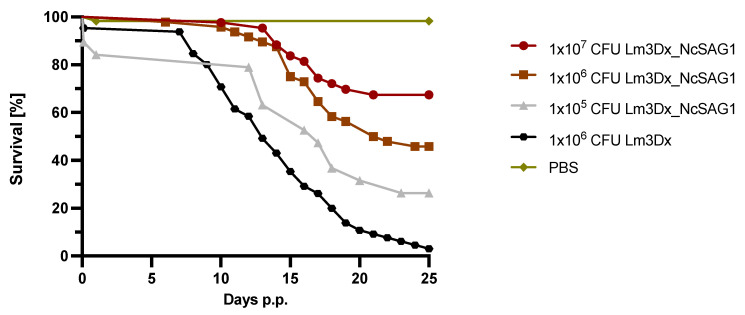
Efficacy assessment of the mutant vaccine strain Lm3Dx_NcSAG1 at different doses in a pregnant neosporosis mouse model. Survival rates of pups were plotted daily in Kaplan–Meier graphs and curves were analyzed by the Log-rank (Mantel-Cox) test. Differences between 1 × 10^7^ CFU Lm3Dx_NcSAG1 and the positive control (C+) curves were highly significant (*p* < 0.0001). Same significant differences were achieved by comparing 1 × 10^6^ CFU Lm3Dx_NcSAG1 survival curves with the curve from the C+ group (*p* < 0.0001). Even the survival curve with the lowest vaccination dose (1 × 10^5^ CFU Lm3Dx_NcSAG1) was statistically different compared to the positive control curve (*p* < 0.0112).

**Figure 3 vaccines-09-01400-f003:**
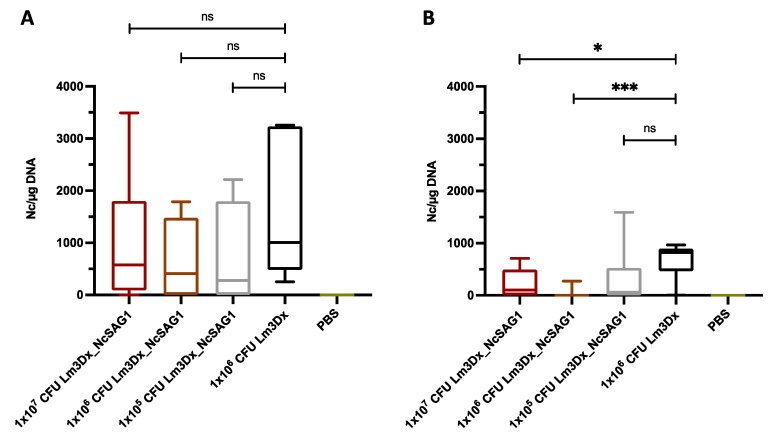
Evaluation of cerebral parasite load of dams (**A**), and non-pregnant mice (**B**) by quantitative RT-qPCR. Brain samples of dams and non-pregnant mice were collected aseptically directly after euthanasia and the cerebral parasite burden was quantified by RT-qPCR. Values are depicted as box plots. No statistically significant reduction of cerebral parasite load was detected between vaccinated and non-vaccinated dams (ns = not significant; Mann–Whitney-*U* test). On the contrary, non-pregnant mice which had been vaccinated with either 1 × 10^7^ CFU or 1 × 10^6^ CFU Lm3Dx_NcSAG1 showed a significant reduction of parasite burden compared to the positive control group receiving the empty vector Lm3Dx (* *p* < 0.0193, *** *p* < 0.0007; Mann–Whitney-*U* test).

**Figure 4 vaccines-09-01400-f004:**
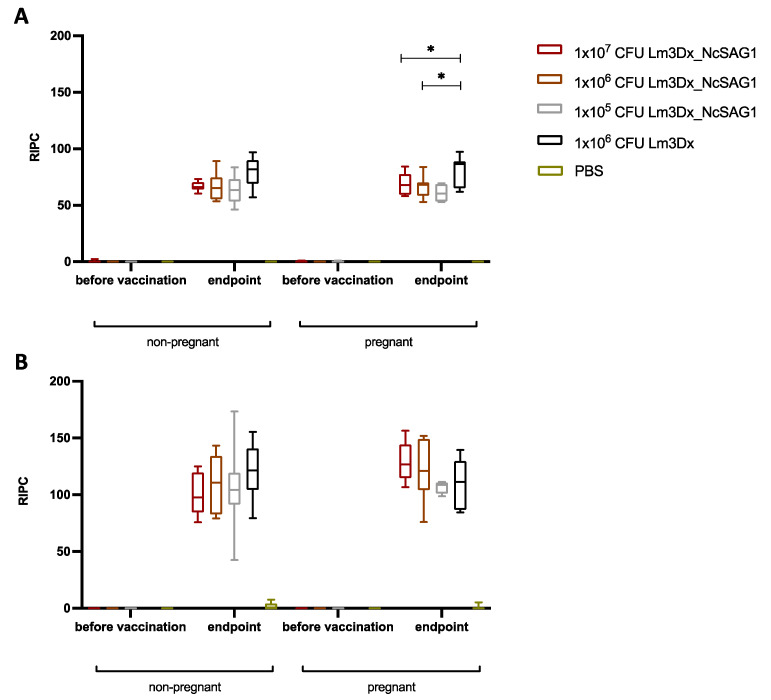
Humoral immune responses (IgG) directed against *N. caninum* extract (**A**) and recombinant NcSAG1 (**B**). Sera were collected prior to vaccination and at the endpoint (38–40 days post challenge) from non-pregnant and pregnant mice. Results are expressed as the mean of RIPC (relative index per cent) compared to the respective positive control that received the empty vector only. (**A**) No significant differences in IgG levels against *N. caninum* were detected between vaccinated non-pregnant mice and the respective positive control, whereas in dams, IgG antibody titers were significantly decreased at the two higher vaccination doses compared to the positive control group (* *p* < 0.0434 for 1 × 10^7^ CFU; * *p* < 0.0343 for 1 × 10^6^ CFU; Mann–Whitney-*U* test) but not between the lowest vaccination dose (1 × 10^5^ CFU) and the positive control. (**B**) Antibody titers directed against recombinant NcSAG1 did not differ significantly in any of the groups.

**Figure 5 vaccines-09-01400-f005:**
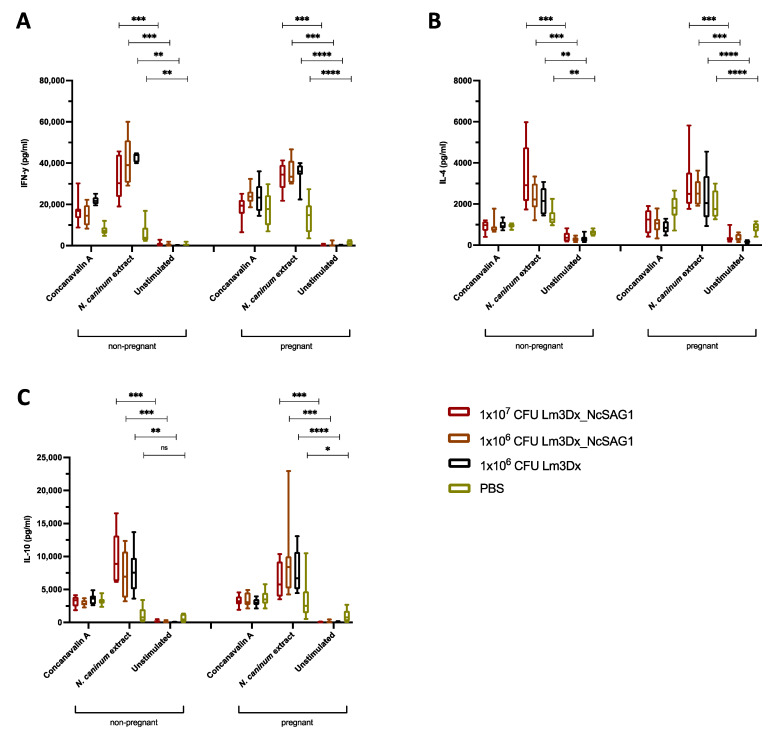
Cytokine measurements after splenocyte re-stimulation. Splenocytes were isolated at the endpoint (38–40 days post challenge) from non-pregnant mice and dams. After isolation, splenocytes were cultured and stimulated for 3 days with either ConA, soluble *N. caninum extract* or remained unstimulated. Results are shown as box plots. (**A**) A strong IFN-γ response was measured in non-pregnant and pregnant mice that were immunized with either Lm3Dx_NcSAG1 or Lm3Dx (empty vector) but not with PBS. (**B**) High levels of IL-4 were observed in mice inoculated with 1 × 10^7^ CFU Lm3Dx_NcSAG1. Decreased IL-4 levels were obtained in mice vaccinated with 1 × 10^6^ CFU Lm3Dx_NcSAG1 and with 1 × 10^6^ CFU Lm3Dx. (**C**) Additionally, high levels of IL-10 were measured after splenocyte re-stimulation with *N. caninum* extract in non-pregnant mice and dams. ***** p* < 0.0001, *** *p* < 0.0002, ** *p* < 0.0022, * *p* < 0.0172, ns = not significant (Mann–Whitney-*U* test).

**Figure 6 vaccines-09-01400-f006:**
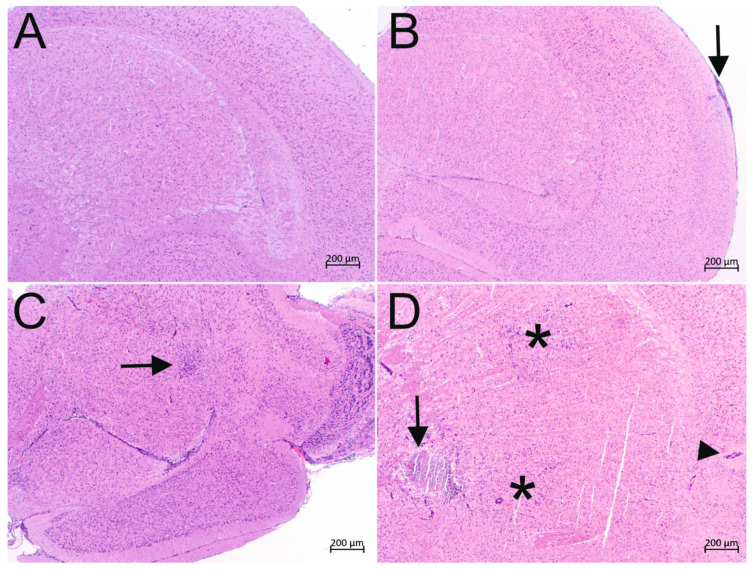
Histology (hematoxylin and eosin-stained paraffin sections) of *N. caninum* infection in brains of animals vaccinated with the Lm3Dx vector. Representative pictures of the Encephalitis grades 0–3 are shown. (**A**) Encephalitis grade 0 in an animal vaccinated with 1 × 10^7^ CFU Lm3Dx_NcSAG1. No brain lesions are visible. (**B**) Encephalitis grade 1 in an animal vaccinated with 1 × 10^7^ CFU Lm3Dx_NcSAG1. A focal mild lymphohistiocytic infiltrate is visible in the meninges (arrow) that slightly extends into the adjacent cortex. (**C**) Encephalitis grade 2 in an animal vaccinated with 1 × 10^5^ CFU Lm3Dx_NcSAG1. A focal-extensive area of marked gliosis and lymphohistiocytic infiltration (arrow) is present in the caudate nucleus. (**D**) Encephalitis grade 3 in an animal inoculated with 10^6^ CFU of the empty vector Lm3Dx. The brain is severely affected by multifocal lesions: Two focal-extensive areas of gliosis and lymphohistiocytic infiltration (asterisks) and a large granuloma with central necrosis (arrow) are present in the caudate nucleus. Additionally, lymphohistiocytic perivascular cuffs (arrowhead) are present in the cortex.

**Figure 7 vaccines-09-01400-f007:**
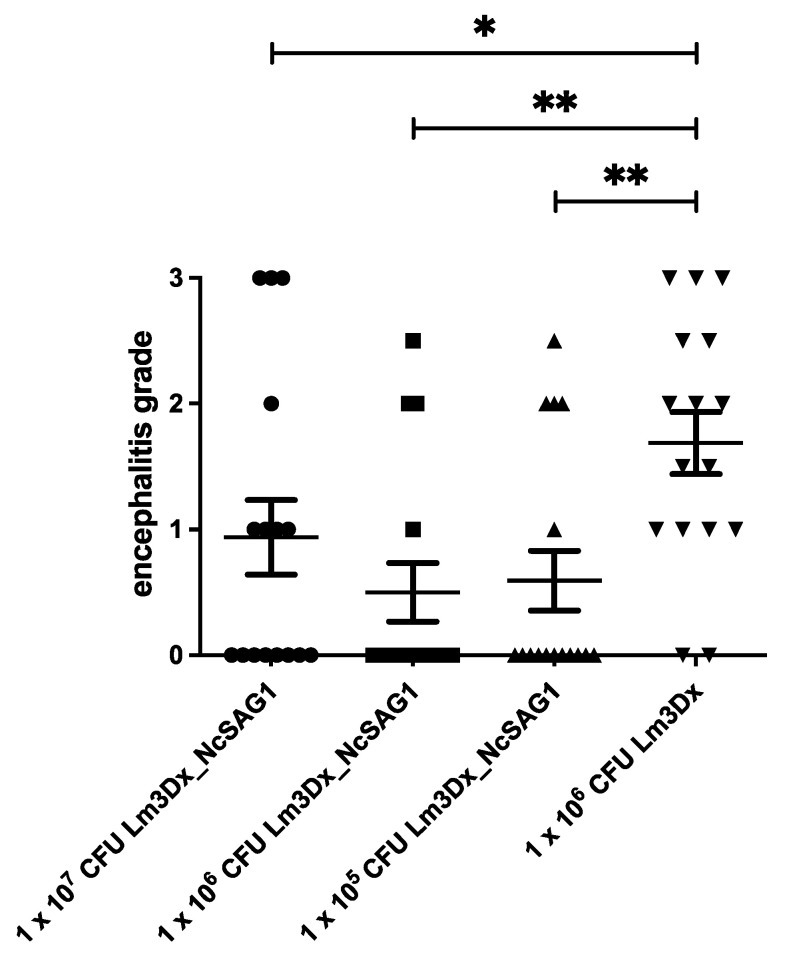
Scoring of encephalitis grades after brain samples were formalin-fixed, paraffin embedded, cut at 4 µm and stained with HE. Cerebral parasite load and associated lesions were assessed by microscopical analysis. Encephalitis grades were analyzed between groups by the non-parametric Kruskal–Wallis test. Every group was compared with the positive control group by using the Mann–Whitney-U test (* *p* < 0.0495 for 1 × 10^7^ CFU Lm3Dx_NcSAG1, ** *p* < 0.0018 for 1 × 10^6^ CFU Lm3Dx_NcSAG1, ** *p* < 0.0034 for 1 × 10^5^ CFU Lm3Dx_NcSAG1).

**Table 1 vaccines-09-01400-t001:** Serology, parasite burden, litter size, neonatal and postnatal mortality rates of *N. caninum* infected mice vaccinated three times with the attenuated mutant *Listeria monocytogenes* strain Lm3Dx_NcSAG1 at different dosages. Animals in the vaccinated groups were compared with the positive control group by the Chi-square (and Fisher’s exact) test.

Vaccination	Challenge	Seropositive for *N. caninum*	*N. caninum* Brain Positive Non-Pregnant Mice	*N. caninum* Brain Positive Dams	Number of Pups per Dam	Neonatal Mortality ^a^	Postnatal Mortality ^b^	*N. caninum* Brain Positive Pups
Lm3Dx_NcSAG1 1 × 10^7^ CFU	1 × 10^5^ NcSp7 tachyzoites	16/16	5/8	7/8	43/8 (∅ 5.4)	0/43 (0%)	14/43 (33%) ^2^	17/43 (39%) ^5^
Lm3Dx_NcSAG1 1 × 10^6^ CFU	1 × 10^5^ NcSp7 tachyzoites	16/16	1/8 ^1^	5/8	48/8 (∅ 6)	0/48 (0%)	26/48 (54%) ^3^	37/48 (77%) ^6^
Lm3Dx_NcSAG1 1 × 10^5^ CFU	1 × 10^5^ NcSp7 tachyzoites	15/15 ^+^	6/11 ^+^	2/4	19/4 (∅ 4.8)	3/19 (16%)	11/16 (69%) ^4^	14/16 (88%)
Lm3Dx 1 × 10^6^ CFU	1 × 10^5^ NcSp7 tachyzoites	16/16	5/6	10/10	65/10 (∅ 6.5)	3/65 (5%)	59/62 (95%)	60/62 (97%)
PBS	1 × 10^5^ dermal fibroblasts	0/16	0/6	0/10	57/10 (∅ 5.7)	1/57 (2%)	0/56 (0%)	0/56 (0%)

^+^ one mouse had to be euthanized after mating due to fighting issues; ^1^
*p* < 0.0256; ^2^
*p* < 0.0001; ^3^
*p* < 0.0001; ^4^
*p* < 0.0077; ^5^
*p* < 0.0001; ^6^
*p* < 0.0021; ^a^ number of pups that were born dead or died within the first two days post-partum; ^b^ number of pups that died between days 3–25 post-partum.

## Data Availability

No publicly archived datasets for this study are available. All data is presented in this article. For further information contact the corresponding authors.

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
