# Peer review of "A Listeria monocytogenes-Based Vaccine Formulation Reduces Vertical Transmission and Leads to Enhanced Pup Survival in a Pregnant Neosporosis Mouse Model"

_vaccines, 2021, doi:10.3390/vaccines9121400_

Round 1

Reviewer 1 Report

Imhof et al developed the attenuated Listeria vaccine that expressing NcSAG1. They showed promising results in the pregnant neosporosis mouse model. Overall, this manuscript was written very well, however, I have some concerns and comments on it.

In Figure 6. They evaluate encephalitis grade, but there is no picture of histology. Please consider putting some representative pictures from each group. I think this is an essential and standard way in this kind of analysis.

Figure 5. They did not show any statistical analysis result in Fig. 5. I think it's a bit complicated. In order to show their vaccine could induce antigen-specific response against Nc extract, they can compare stimulation to unstimulation with Mann-Whitney-U test.

Figure 2. I see “probability of survival [%]”. I think this might be just “survival [%]”. I recommend they add “of pups” in legend, i.e. Survival rates “of pups” were plotted daily...

IFN-y should read IFN-γ "gamma".

No explanation of †, on Imhof and Pownall, I think its co-1st author.

Author Response

We thank the reviewer for constructive comments on our manuscript. Please find our responses below.

Imhof et al developed the attenuated Listeria vaccine that expressing NcSAG1. They showed promising results in the pregnant neosporosis mouse model. Overall, this manuscript was written very well, however, I have some concerns and comments on it.

In Figure 6. They evaluate encephalitis grade, but there is no picture of histology. Please consider putting some representative pictures from each group. I think this is an essential and standard way in this kind of analysis.

Response: we added an additional figure depicting representative images of histological sections as requested by the reviewer.

Figure 5. They did not show any statistical analysis result in Fig. 5. I think it's a bit complicated. In order to show their vaccine could induce antigen-specific response against Nc extract, they can compare stimulation to unstimulation with Mann-Whitney-U test.

Response: this was done and the corresponding figure 5 and figure legend were revised accordingly

Figure 2. I see “probability of survival [%]”. I think this might be just “survival [%]”. I recommend they add “of pups” in legend, i.e. Survival rates “of pups” were plotted daily...

Response: done as requested.

IFN-y should read IFN-γ "gamma".

Response: correct, and changed accordingly

No explanation of †, on Imhof and Pownall, I think its co-1st author.

Response: correct remark, this indicates common first authorships, and is now indicated.

Reviewer 2 Report

In order it can be understand all the vaccination experiments is necessary to add to the section 2.6 what it consist the serum positive control. Moreover, during discussion is always mentioned the data compared with the positive control group that is not described, in what it consisting?  

Author Response

We thank the reviewer for his comments on our manuscript. Please find our response below.

In order it can be understand all the vaccination experiments is necessary to add to the section 2.6 what it consist the serum positive control. Moreover, during discussion is always mentioned the data compared with the positive control group that is not described, in what it consisting?  

Actually, the positive and negative controls are described in M&M section in lanes 151-160. We have now also included this information on the two control groups in lanes 207-208.

Round 2

Reviewer 1 Report

Thanks for responding to my request.